# Conceptual Model on Access to Food in the Favela Food Environment

**DOI:** 10.3390/ijerph21111422

**Published:** 2024-10-26

**Authors:** Luana Rocha, Daniela Canella, Raquel Canuto, Mariana Jardim, Letícia Cardoso, Amelia Friche, Larissa Mendes

**Affiliations:** 1Department of Preventive and Social Medicine, Faculty of Medicine, Federal University of Minas Gerais, Belo Horizonte 30130-100, MG, Brazil; 2Department of Applied Nutrition, Institute of Nutrition, Rio de Janeiro State University, Rio de Janeiro 20550-013, RJ, Brazil; danicanella@gmail.com; 3Department of Nutrition, Faculty of Medicine, Federal University of Rio Grande do Sul, Porto Alegre 90035-003, RS, Brazil; raquelcanuto@gmail.com; 4Pediatric Department, Faculty of Medicine, Federal University of Minas Gerais, Belo Horizonte 31270-901, MG, Brazil; zogbij@gmail.com; 5Sergio Arouca National School of Public Health, Oswaldo Cruz Foundation, Rio de Janeiro 21040-900, RJ, Brazil; leticiadeoliveiracardoso@gmail.com; 6Department of Speech Pathology, Faculty of Medicine, Federal University of Minas Gerais, Belo Horizonte 31270-901, MG, Brazil; gutafriche@gmail.com; 7Department of Nutrition, School of Nursing, Federal University of Minas Gerais, Belo Horizonte 31270-901, MG, Brazil; larissa.mendesloures@gmail.com

**Keywords:** conceptual model, favela, food environment, food access

## Abstract

The inequalities of Brazilian society are amplified in favelas, affecting access to basic sanitation, health, education services, and food. More research is needed to better understand the favela food environment and propose appropriate public food and nutrition policies to increase the availability of and access to healthy food. In this context, this study aimed to develop a conceptual model of the relationship between access to food and the favela food environment and its determinants. In developing the conceptual model, this study undertook a bibliographical survey of the food environment, and a preliminary version was submitted to an expert panel. The model represents a set of dimensions (individual, micro-environment, macro-environment, and decision-making) and elements that interact in a complex manner and help understand access to food in areas subject to multiple social vulnerabilities. This model can guide future research and aid policymakers in designing effective strategies to improve the food security and health of populations in areas of high socio-spatial vulnerability.

## 1. Introduction

Favelas are defined by the Brazilian Institute of Geography and Statistics as urban areas with a predominance of households presenting different levels of legal insecurity and at least one of the following criteria: lack or incomplete provision of public services, predominance of buildings, road planning, and infrastructure often self-generated or guided by urban planning and construction parameters different from those established by public bodies, and location in an area with occupation restrictions determined by environmental or urban planning legislation [1]. More broadly, poor and populated areas of a city can also be considered favelas [2].

The construction of favelas—or their self-construction—is marked by strategies to ensure the right of access to the city, as a response to processes of spatial segregation of social minorities, vulnerable populations, or populations at social risk. The response also includes programs to remove dwellers (for the construction of modern and, consequently, more expensive housing) and migration from the countryside to the city, resulting in the social occupation of hills and empty spaces around urban centers and on the outskirts of cities [3,4,5]. Without government support, these locations of urban agglomeration lack access to public services and basic sanitation, presenting diverse construction profiles and illegal land occupation [3,4,5]. These characteristics have been perpetuated for decades and continue to exist in modern society. Favelas are marked by unequal access to several basic services and human rights, including access to food.

Rocha et al. [6] evaluated the perception of Brazilian favela dwellers about the food environment in their neighborhoods, reporting that their access to food is permeated by a lack of information about food, insufficient income, and low availability of stores that sell fresh and minimally processed food (healthy food) at affordable prices. The food environment refers to the physical, economic, political, and socio-cultural factors that influence access and availability of food in a territory [7]. Secondary data from a Brazilian metropolis show that ultra-processed food stores (snack bars, bars, and candy stores) are widely available in favelas, while those selling healthy foods (fish markets, farm produce, and butchers and meat shops) are fewer and more distant [8]. In favelas, the food environment and access to food result from a complex territorial interplay, which is affected by the stereotype perpetuated by the general population that favelas are dangerous places and a hotbed of crime [9]. Therefore, research on the favela food environment must consider the context in which the favela population lives.

Food environment researchers have focused on low- and middle-income countries to create conceptual models to provide elements that help organize and interpret complex information about food environment constituents and determinants [10,11]. However, these models do not consider the complexity of environments characterized by socio-spatial inequalities, such as favelas, lacking specificity. Mendes et al. [12] mapped published studies on the food environment in Brazil, reporting that conceptual models did not support most studies (77.7%). This scenario may be due to the lack of specific models on the food environment, which have failed to consider territorial particularities for use in different countries and contexts.

Therefore, we should better understand the food environment of populations exposed to spatial and social vulnerabilities—such as the favelas—to propose public policies on food, nutrition, health, and food and nutritional security aimed at increasing their access to healthy food and ensuring the human right to adequate food. This study aims to develop a conceptual model of the relationship between access to food and the favela food environment and its determinants, with the goal of supporting new research on food environments in vulnerable regions at social risk.

## 2. Materials and Methods

The conceptual model was constructed using the food environment concept proposed by Downs et al. [10] and the method proposed by Souza Filho and Struchiner [13], which includes the following stages: 1. identifying and delimiting the object of study; 2. cognitive retrieving and brainstorming; 3. representing the conceptual model; 4. reviewing the literature on the subject; 5. structuring the conceptual model; 6. submitting the conceptual model to experts; and 7. restructuring and finalizing the conceptual model.

We chose the favela food environment as the study object (stage 1). We referred to the food environment concept proposed by Downs et al. [10] in this study because it considers the dynamics of food environments in middle-income countries, such as Brazil, including the informal environment as part of constructed spaces. According to Downs et al. [10], the food environment is the consumer’s interface with the food system, which encompasses the availability, ease of use, convenience, promotion, quality, and sustainability of foods and beverages in the formal or informal wild, cultivated, and built spaces that are influenced by the sociocultural and political environment and the ecosystems in which they are located (stage 2).

Stage 2 is related to the process of constructing the conceptual model. Considering that the favela food environment is rarely studied in the literature, we created focus groups with Brazilian favela dwellers to assess their perception of the food environment in their neighborhoods and its relationship with food access and consumption, detailed by Rocha et al. [6]. Two online focus groups were held, with five participants living in Brazilian favelas in each group. Using the snowball sampling technique, participants were invited through social networks and contact with community leaders and non-governmental organizations working in favelas [6]. Then, we referred to Rocha et al.’s qualitative study with Brazilian favela dwellers on access to food [6] and their study on the distribution of and access to formal food stores in Belo Horizonte Favelas [8]. Formal food stores are usually registered with government institutions and are regulated by legal measures. Specifically, Rocha et al. [8] analyzed secondary data on food stores in Belo Horizonte (MG, Brazil) favelas to verify the distribution of formal food stores by type and to characterize the physical access to these stores in these neighborhoods [8]. These two studies, which are associated with the model of Downs et al. [10], informed the model-construction process to develop a first representation of the conceptual model (stage 3).

Stage 4 comprised a bibliographic survey on food environment models and research on the Brazilian food environment about favelas in PubMed, Scopus, Web of Science, and Scielo, without restricting the language. Based on the bibliographic survey and the results of the earlier stages, a preliminary conceptual model on the relationship between the favela food environment and access to food (stage 5) was proposed. This model consists of a graphic diagram and a table detailing its elements (Appendix A). The preliminary version of the conceptual model includes the main factors influencing access to food by Brazilian favela dwellers at individual, micro-environment, macro-environment, and decision-making levels.

An expert panel evaluated the first version of the conceptual model between October and December 2023 (stage 6). The literature recommends the participation of 5–20 professionals in this stage [14,15], which was structured into four phases. The first phase was identifying the experts to be invited; in the second, they were sent an e-mail with an introduction to the study and an invitation to participate; the third phase involved evaluating the model using an online form; and in the fourth phase, the suggestions were analyzed. Appendix A systematizes the suggestions with respective justifications for including them in or excluding them from the final version of the model. The online form used in the third phase contained closed questions asking experts to indicate their level of agreement with the dimensions and elements of the model. It also provided space for them to write any questions and suggest improvements (Appendix A).

The study included 25 experts: 7 members of civil society who live and work in favelas, and 18 researchers and university professors with extensive experience in the food environment, epidemiology and public health, and health inequalities. The experts invited to make up the panel were nationally recognized researchers in the areas of study mentioned above and civil society actors who live and work in Brazilian favelas. The panel had one participant from the North of Brazil, four from the Northeast, six from the South, thirteen from the Southeast, and one from the Center-west. In total, there were 3 men and 22 women, of whom 11 were White and 14 Black. Relevant elements indicated as suggestions were incorporated into the final version of the conceptual model.

The degree of agreement was quantified using a five-point scale ranging from “strongly agree” to “strongly disagree”, which evaluated the conceptual model dimensions, elements, concepts, and graphic representation. These results were used to calculate the percentage of agreement between experts, with 90% considered acceptable [16]. The following formula was used:Percentual of agreement (%) = ((number of experts agreeing)/(total number of experts)) × 100

Those who responded with “strongly agree” or “partially agree” were classified as agreeing (numerator in the formula).

The final version of the conceptual model (Appendix A) was improved by a graphic designer and is presented in the Section 3 (stage 7).

### Ethical Aspects

This study was approved by the Research Ethics Committee of the Federal University of Minas Gerais (CAAE: 54588221.7.0000.5149 and 48190221.2.0000.5149). All participants signed a digital Informed Consent Form to participate in the study.

## 3. Results

We considered the five dimensions of food access defined by Penchansky and Thomas [17], which were updated by Caspi et al. [18] in their study on the outcome of access to food: availability, physical proximity/accessibility, financial accessibility, acceptability, and convenience. Caspi et al. [18] evaluated the same outcome as the conceptual model proposed in this study. The final version of the model is depicted graphically in Figure 1 and Figure 2.

The dimensions of the model were built considering the individual, micro-environment, macro-environment, and decision-making levels (Figure 1). The individual level was defined based on the concept of individual-level factors related to food choices and behaviors by Story et al. [19]. This level refers to individual characteristics, conditions, and behaviors that can affect and influence access to food.

The micro-environment was defined based on Swinburn et al. [20]. It refers to a context in which groups of people come together for specific purposes that involve or affect food. These contexts are geographically distinct, relatively small, and potentially influenced by individuals, such as the region where they live, their work or study place, church, temple, or other place where they can access food. The macro-environment was also defined based on Swinburn et al. [20]. It refers to a group of industries, services, or support infrastructure that influences food availability and consumption in various contexts within the micro-environment.

The decision-making level was based on the concept proposed by Castro and Canella [21], who developed a conceptual model for the organizational food environment. In this context, the decision-making level refers to how power relationships and decision-making processes affect the food environment through municipal, state, or national policies, programs, laws, and regulations. This level refers to the legal tools that influence the environment, enabling actions in the service structures described in the macro- and micro-environment. It also includes the legal tools that influence food availability in the micro-environment and can affect access to food at the individual level (e.g., social policies that affect the income of favela dwellers or improve financial access to food).

Figure 2 shows the complete graphic representation of the conceptual model on food access in favelas, describing the elements included in each dimension.

Table 1 shows the definition of each element included in the dimensions of the conceptual model.

### Percentage of Agreement

Table 2 shows the percentage of agreement for each aspect evaluated by the expert panel and provides all the dimensions and parameters that should be included in a favela food environment model. The highest percentage of general agreement was 100%, and the lowest was 92%. The categorization by members of civil society living and working in favelas and by researchers and professors showed that the highest percentage of agreement was 100% for both types of experts, and the lowest percentages were 80% and 93.33%, respectively.

## 4. Discussion

The conceptual model presented in this study systematizes access to food in the context of the favela food environment. The dimensions evaluated represent the influence of each element on access to food by favela dwellers, as they coexist and interact with each other and with the territory, highlighting the complexity of these areas.

The conceptual model has four dimensions. The individual level refers to individual characteristics that affect access to food. The micro-environment level refers to the characteristics of the territories in which individuals live. These characteristics directly interfere with access to food but are also mediated by individual characteristics. The macro-environment level refers to structural issues in society and the environment in which individuals live. These issues affect the services offered in the neighborhoods and the way society acts and thinks. The decision-making level refers to government policies, programs, and actions that influence all levels explained in the conceptual model. These decisions directly impact access to food and related aspects that measure access to food by the population.

The social determinants of health (SDH) are essential at all levels of the model, being crucial in the discussion on access to food in geographical areas resulting from social inequalities, such as favelas. The World Health Organization highlights that SDH encompasses the conditions in which people are born, grow, work, live, and age, as well as the broader set of forces and systems that shape these conditions, including economic policies and systems, development agendas, social norms, social policies, and political systems [22]. In this context, according to the SDH model proposed by Solar and Irwin [23], the food environment can be considered an intermediate determinant of health. It is influenced by political, economic, and social conditions and their inequities, simultaneously being a determining factor in the health of individuals [24,25,26,27,28]. Thus, understanding its dynamics is essential to capture the complex context in which people live and how it can affect the population’s health [29,30].

The elements in the conceptual model extend beyond the food availability dimension. They consider factors that can interfere with access to food in favelas and areas of high social vulnerability, such as security and violence, which deter and discourage food vendors and hinder movement and access to food within the territory. These elements have not been characterized as good or bad or added specific characteristics (e.g., healthy food store) to adapt the model to different realities and contexts in favelas and high vulnerability regions, which can vary by country, city, and even by the urban history of the territory. Thus, one or more elements may not be included in the study of specific territories to adapt the model to the local reality (e.g., when studying a favela with no community gardens, this element would not be included).

We emphasize that the construction of the model, including the methodological and didactic choice of separating the elements by levels, was based on current studies on the favela food environment [6,8] and on conceptual model approaches to the food environment in Latin America or low- and middle-income countries [10,21,24]. The expert panel also considered the participation of members of civil society living and working in favelas to validate the content of the conceptual model with people who experience this reality. All dimensions, elements, and concepts were well-evaluated and considered important by the experts. However, civil society experts found the decision-making level difficult to understand, and professors and researchers had several questions about it. Castro and Canella [21] first described the decision-making level when they developed a conceptual model of organizational food environments. This level refers to the governance of the environment. As for access to food, government policies and programs should be included, and their impact should be studied to improve access to food in the territories. We opted to keep the decision-making level, adding examples and a more detailed definition.

Over the last three decades, researchers have sought to understand the dynamics of food environments in their respective territories [10,11,19,24,31,32,33], but many conceptual models have been produced within or by authors from the Global North, despite their attempts to explain food environments broadly or in middle- and low-income countries [10,11]. The specificity of low- and middle-income countries was explored by Gálvez-Espinoza et al. [24], who presented a conceptual model to systematize the factors that condition the food environments of the Chilean population. Turner et al. [11] tried to develop a broad conceptual model for low- and middle-income countries, but the lack of a broad review of studies from these countries resulted in a limited view of scientific production and conclusions that were not consistent with reality. In 2020, Downs et al. [10] coined a new broad proposal for the term food environments that could be applied to countries of all income strata, thus expanding the possibility of applying the conceptual model to low- and middle-income countries mainly by including informal trade as one of the dimensions of the constructed food environment [10]. The informal food environment refers to informal food sales by a store (street vendors, kiosks, etc.) that may or may not be regulated and supervised by government bodies, is characterized by a lack of specialization, low capital investment, lack of accountability, non-payment of some or all taxes, and limited social innovations [10,34]. In middle- and low-income countries, informality is also linked to the country’s food culture, providing income for a significant portion of vulnerable groups and being present in most vulnerable territories, including favelas [34].

Despite being fundamental, broad conceptual models may not fully capture the complexity of specific contexts found in low- and middle-income countries, such as regions of high social vulnerability. For example, Ambikapathi et al. [35] presented a specific model for the informal food environment in the city of Dar es Salaam, Tanzania, which helped better understand the population’s relationship with food purchases in the territory.

Thus, the conceptual model presented in this study included elements specific to the context of favelas to elucidate the factors influencing access to food in these areas. This approach aimed to address the specificities of vulnerable territories due to their geospatial location and promote the study of favela food environments. Our model was developed within the context of favelas, but it could also be applied to the context of cities because we carefully identified elements that explain social vulnerability in both individual and territorial contexts, transcending issues related to the city and urban environment. This model is crucial for Global South countries, where the formation of territories like favelas is prevalent, expanding possibilities for studying the food environment and its determinants in low-income regions.

A limitation of this study is that the literature review was not systematized according to an internationally recommended protocol, such as the Preferred Reporting Items for Systematic Review and Meta-Analysis. However, the literature on food environments was updated through regular searches in bibliographic databases during the development of the conceptual model. Although the expert panel was restricted to Brazil, we believe the model can be used in other countries.

Finally, the conceptual model presented in this study is a tool for understanding the complex relationship between the favela food environment and access to food. The model includes several dimensions and elements that reflect the reality of this territory, providing a comprehensive view of the influences on food access and highlighting the importance of policies and interventions adapted to local specificities. This model can guide researchers and policymakers in conducting studies and developing effective strategies aimed at improving food security and health outcomes for populations in areas of high social vulnerability.

## 5. Conclusions

This study proposes an innovative conceptual model for understanding access to food in the favela environment, focusing on the factors that influence access in these territories. The model represents a set of dimensions and elements that interact in a complex manner and helps understand access to food in vulnerable areas whose inhabitants are at social risk. Although it was developed for the Brazilian context, the conceptual model can be used by researchers and students in other countries with geographical areas sharing similar income levels and socioeconomic vulnerabilities. Its comprehensive list of elements broadly evaluates the territory with a focus on social inequities and situations of vulnerability. In addition, this model is focused on the food environment, an area often neglected by public policies and the literature, drawing attention to vulnerable territories and promoting the development of new research. This can serve as a basis for further research to develop effective public policies and programs aimed at favelas.

## Figures and Tables

**Figure 1 ijerph-21-01422-f001:**
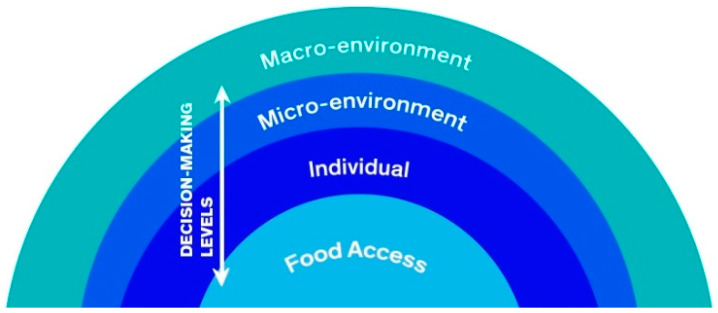
Dimensions considered in the conceptual model on access to food in favelas. Note: The conceptual model should be considered in the context of the social determinants of health, which include the food environment.

**Figure 2 ijerph-21-01422-f002:**
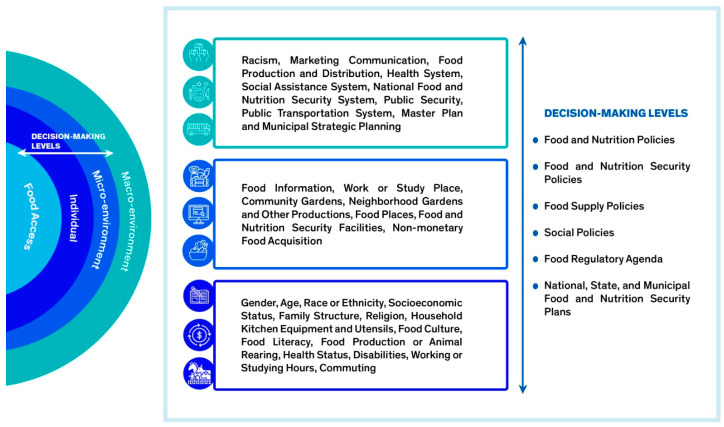
Elements considered in the conceptual model on access to food in favelas.

**Table 1 ijerph-21-01422-t001:** Definitions of the elements included in the conceptual model on access to food in favelas.

Dimension	Elements	Definition
Individual	Gender	Characteristics, roles, behaviors, expectations, and socially constructed identities associated with gender, being a man, being a woman, or other gender identities such as transgender, non-binary, or agender [1]. Gender identity also relates to power relationships between its different forms of expression. Examples: cisgender woman, transgender woman, cisgender man, transgender man, and agender.
	Age	Stage of life in which an individual is measured from the time of birth. Examples include childhood, adolescence, and senescence.
	Socioeconomic status	The amount of money a person receives periodically as remuneration for work or services (e.g., salary, pension, allowance) and other government benefits and considers the individual’s education level.
	Race or ethnicity	Race is the social construction and categorization of people based on perceived physical traits that maintain a socio-political hierarchy [2]. Ethnicity refers to the characterization of people based on shared culture, ancestry, and history [3].
	Family structure	The number of individuals living together and the composition of the household (e.g., a family composed only of women or men, the presence of children and adolescents, and other family compositions).
	Religion	A set of beliefs and social practices related to the notion of the sacred, uniting all adherents into the same moral community [4]. Some practices involve temporarily or permanently consuming or banning certain foods, or the community can help by donating and sharing food.
	Food culture	A set of ideas, beliefs, knowledge, and practices related to food and shared between and within groups over time [5].
	Working or studying hours	The daily period when the dweller is available for work or study. It includes commuting time for in-person activities.
Individual	Health status	Dwellers’ health conditions that can affect access to food due to restrictions, e.g., comorbidities that affect mobility or daily activities (obesity, chronic non-communicable diseases).
	Disabilities	The presence of disabilities that can affect access to food due to restrictions, such as congenital or acquired physical or mental incapacities.
	Food production or animal rearing	Producing food or raising animals for personal consumption, with the possibility of selling or exchanging surplus locally. Examples include backyard or balcony vegetable gardens (vertical or horizontal) and raising animals for consumption and sale, like milk, honey, and eggs.
	Food literacy	The dweller’s ability to read, understand, and judge the quality of nutritional information; to seek and exchange knowledge about food and nutrition; to buy and prepare food; to reflect critically on the factors involved in personal food choices; and to understand the impact of these choices on society [6].
	Household kitchen equipment and utensils	It considers the materials, equipment, and utensils available in the household to store and prepare food.
	Commute	Time spent and distance traveled between different physical points using any means of transport (own, alternative, or public). It reflects the individual’s routine of commuting around the city.
Micro-environment	Food Information	Food information available in the environment, such as nutritional tables and ingredient lists, food advertising, or nutritional claims.
	Community and neighborhood vegetable gardens and other local food production sites	Presence of community or neighborhood gardens, productive backyard farming, or other forms of food production or animal rearing for free distribution, exchange, and sale within the community. Examples include private vertical or horizontal backyard or balcony gardens and community gardens initiated either by the population or the government in public spaces.
	Food fairs	Physical spaces for selling food in permanent locations using tents and collapsible equipment to provide fresh or minimally processed food, and culinary preparations, among other types of food.
Micro-environment	Non-monetary food acquisition	The exchange of food items without using currency, or small or large food donation by a single person, a group of volunteers, or organizations and institutions, which may be mediated by non-profit organizations or community leaders. This food can be donated by churches, non-governmental organizations, government institutions, or other associations or individuals.
	Formal food stores	Formal fixed-location food stores that pay taxes and are inspected and registered by government bodies. These stores are regulated through state, national, or other local registrations, such as unions.
	Informal food stores	Informal fixed-location food stores that may or may not undergo inspection by health authorities but that pay no taxes. These stores are characterized by a lack of specialization, low capital investment, lack of accountability, non-payment of some or all taxes, and limited social innovations, such as new forms of payment (barter system), or selling in unconventional places [7]. The food stores are not regulated by state, national, or other registrations. Examples include selling food in garages or doorsteps or having a sign indicating they sell various culinary preparations.
	Food and nutrition security facilities	Physical structures or spaces that provide public services aimed at promoting access to quality food, adequate nutrition, and food and nutritional security for the population. Examples include popular restaurants, food markets, food banks, community kitchens, and solidarity kitchens.
	Street vending	Street vending can take place in open or closed spaces, whether with a fixed location or not. Street vendors may or may not be registered with government bodies and pay taxes, but they are subject to inspection (e.g., municipal street vendor registers). Examples include food vendors who have a sales space that can be moved around (carts, mobile stands, cars, bicycles, etc.).
	Food e-commerce	Selling food through online stores, with the entire product purchase being online: product selection, determination of an address for delivery or pick-up, payment, and purchase.
Micro-environment	Online food delivery platforms	Applications that host one or more companies that sell both ready-to-eat or non-ready-to-eat food items with a delivery service. These business platforms provide order, delivery, and payment services to food stores.
	Local non-profit organizations and institutions	Non-profit organizations that provide free support and services to favela dwellers. Examples include social movements, dwellers’ associations, and the Central Única de Favelas in Brazil [8].
	Other organizations	For-profit organizations that are open to working in the favelas or have some kind of influence there. Examples include good industries (which can promote food produced in favelas) and criminal organizations (criminal factions and militias).
	Social services	These include any type of public or private social assistance service or unit that has direct contact with the individual (e.g., Social Assistance Reference Center, Coexistence and Bond Strengthening Service).
	Health services	These include any type of public or private health service or unit that has direct contact with the individual (e.g., basic health units, private practices).
	Internet access	Availability of broadband connection or internet access points through private of free wireless networks.
	Security and violence	Perception of security and violence experienced by dwellers and food vendors influenced by the presence of drug trafficking, police violence, confrontations, assaults, criminal factions, militias, and other types of criminal organizations that can influence their sense of security or insecurity.
	Alternative transportation	Unregulated collective or individual means of transportation for commuting within the favela. Examples include motorcycle taxis and vans.
	Public transportation	Accessible public transportation that accesses favelas, travel within this territory, and connect it to other points in the city.
	Work or study place	Food sold and/or provided free of charge for immediate consumption in educational institutions and workplaces located in favelas or frequented by favela dwellers.
Micro-environment	Basic sanitation	Access to basic services such as drinking water distribution, sewage collection and treatment, urban drainage, and solid waste collection.
	Walkability	The physical structure of the urban space that allows and favors walking [9]. Examples include street elevation level; presence, quality, and size of sidewalks; obstructions; street lighting; presence of trees; crosswalks; presence and access to bus stops; and size of blocks.
	Social cohesion	The sense of unity and level of interaction among community members for a common purpose [10].
	Social capital	The network of social relations, norms, trust, and cooperation within the community. It includes elements of social organization and civic engagement and networks of associations aimed at achieving a common good or purpose [11].
Macro-environment	Racism	Racism is a form of discrimination that considers race or ethnicity as the basis for practices that result in advantages or disadvantages for individuals based on their group affiliation [12]. In Brazil, racism manifests through the accumulation of privileges by White individuals at the expense of Black individuals. It is expressed through actions, beliefs, and political systems operating across different levels, from internalized to interpersonal, structural, and systemic.
	Marketing communication	The set of marketing strategies, messages, and practices used by companies and organizations in the food industry to promote and advertise their food products. These strategies aim to influence consumer perception, increase brand awareness, and stimulate the purchase of food products, being mostly used by the ultra-processed food industry. Examples include television and social media advertising, attractive labels with health claims, reward programs, value combinations, and discounts.
	Master plan and municipal strategic planning	Urban planning and management tools used by cities and authorities to guide urban development and planning.
Macro-environment	Food production and distribution	Interconnected processes, activities, and infrastructures involved in food production and distribution. They encompass food cultivation, the transformation of raw materials into food products, and the physical movement of food from production to points of sale or consumers. Examples include the ultra-processed food industry, the grain processing industry, and family farmers’ cooperatives.
	Health system	Considers the infrastructure and the set of health services organizing activities in the territory.
	Social assistance system	Considers social protection services for individuals.
	National food and nutrition security system	A structure that puts the goals of the food and nutrition security policy into practice.
	Public security	Strategies, guidelines, actions, and measures implemented by governments and public institutions to promote security, prevent crime, protect citizens, and ensure compliance with the law in each district. These actions are designed to address security-related challenges ranging from crime prevention to emergency and disaster response.
	Public transportation system	Infrastructure and services organized to provide efficient and accessible movement of people within a city, metropolitan region, or urban area. It is designed to meet the mobility needs of urban populations.
Decision-making	Food and nutrition policies	Strategies, plans, programs, and actions implemented by governments and organizations to promote healthy eating and ensure adequate and sufficient access to healthy food.
	Food and nutrition security policies	Strategies, plans, programs, and actions implemented by governments and organizations to promote food and nutrition security. These policies are designed to address issues related to nutrition and access to healthy and sustainable food.
	Social policies	Actions, programs, measures, and strategies implemented by the state or other governmental and non-governmental institutions to address social issues and promote the population’s well-being. These policies aim to meet basic needs and ensure the fundamental rights of people, especially those in situations of vulnerability or social disadvantage.
Decision-making	Food supply policies	Strategies, regulations, government actions, and programs aimed at ensuring adequate and sustainable food supply for the population of a country, region, or community. The main aim of these policies is to ensure access to safe, nutritious, and sufficient food to meet the basic dietary needs of the population.
	Food regulatory agenda	Projects and normative tools aiming to regulate food-related activities, such as labeling rules, food advertising, food tax rules, and other regulatory provisions on food items.
	National, state, and municipal food and nutrition security plans	Actions implemented by food production and distribution structures to improve access to healthy food in sufficient quantity and quality in the territories.

[1] Organizacão Pan-Americana de Saúde (OPAS), Equidade de gênero em saúde, https://www.paho.org/pt/topicos/equidade-genero-em-saude#:~:text=O%20g%C3%AAnero%20se%20refere%20%C3%A0s,mudar%20ao%20longo%20do%20tempo, 2024 (accessed 20 January 2024); [2] APA Dictionary of Psychology, Race, https://dictionary.apa.org/race, 2023 (accessed 20 January 2024); [3] APA Dictionary of Psychology, Ethnicity, https://dictionary.apa.org/ethnicity, 2023 (accessed 20 January 2024); [4] E. Durkheim, As Formas Elementares da Vida Religiosa, Trad. Paulo Neves, São Paulo, Martins Fontes, 1996; [5] U.P. Verthein, L. Amparo-Santos, A noção de cultura alimentar em ações de educação alimentar e nutricional em escolas brasileiras: uma análise crítica, Ciênc. saúde coletiva 26 (2021) 4849–4858; [6] C. Krause, K. Sommerhalder, S. Beer-Borst, T. Abel, Just a subtle difference? findings from a systematic review on definitions of nutrition literacy and food literacy, Health Promotion International 33 (2019) 378–389; [7] Food and Agriculture Organization (FAO), Promessas e desafios do setor informal de alimentos em países em desenvolvimento Roma, 2011; [8] Central Única das Favelas (Cufa), Sobre a Cufa, https://cufa.org.br/quem-somos/2024, (accessed 20 January 2024); [9] Instituto de Políticas de Transporte and Desenvolvimento (ITDP), Índice de Caminhabilidade versãp 2.0 Ferramenta, 2019; [10] S. Karuppannan, A. Sivam, Social sustainability and neighbourhood design: An investigation of residents’ satisfaction in Delhi, Local Environment 16 (2011) 849–870; [11] I. Kawachi, B.P. Kennedy, R. Glass, Social capital and self-rated health: A contextual analysis, American Journal of Public Health 89 (1999) 1187–1193; [12] Brasil, Presidência da República Casa Civil, Decreto nº 65.810, de 8 de dezembro de 1969, Promulga a Convenção Internacional sôbre a Eliminação de tôdas as Formas de Discriminação Racial, Diário Oficial da União, 1969.

**Table 2 ijerph-21-01422-t002:** Percentage of agreement for each aspect evaluated by the expert panel.

Aspect Analyzed	Percentage of General Agreement	Percentage of Agreement Among Members of Civil Society	Percentage of Agreement Among Researchers and Professors
All constituent elements included are relevant to the favela food environment	100%	100%	100%
The construction blocks included in the model cover all the components relevant to the favela food environment	96%	90%	100%
The conceptual model is clear	96%	90%	100%
The graphical representation of the conceptual model is clear	96%	90%	100%
The terms naming the constituent elements of the graphical representation of the conceptual model are clear	96%	90%	100%
The description of access to food is easy to understand	100%	100%	100%
All the dimensions included are relevant to the favela food environment	100%	100%	100%
All the dimensions that should be included in a model for the favela food environment are included in the table	92%	80%	100%
The dimensions are easy to understand	100%	100%	100%
The description of the dimensions is easy to understand	100%	100%	100%
All the parameters included are relevant to the favela food environment	100%	100%	100%
All the parameters that should be included in a model for the favela food environment are included in the table	96%	100%	93.33%
The variables are easy to understand	96%	90%	100%
The description of the variables is easy to understand	96%	90%	100%

## Data Availability

All data are available in the article and Appendix A.

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
