# Peer review of "Conceptual Model on Access to Food in the Favela Food Environment"

_ijerph, 2024, doi:10.3390/ijerph21111422_

Round 1

Reviewer 1 Report

Comments and Suggestions for Authors

Choosing to study the favelas is unique and novel. The manuscript is well-written and easy to follow. I am unsure why Racism is included in the Macro-environment and how it affects food access. Race or ethnicity is included at the individual level and captures the demographic characteristics of favela residents. 

As a follow-up study, employing ArcGIS to map and identify the favela's food environment would be interesting. 

Author Response

Comments

Response

Choosing to study the favelas is unique and novel. The manuscript is well-written and easy to follow. I am unsure why Racism is included in the Macro-environment and how it affects food access. Race or ethnicity is included at the individual level and captures the demographic characteristics of favela residents.

Thank you for your thoughtful feedback. We included racism in the macroenvironment dimension of the model because of its importance in shaping food access for black people. While race or ethnicity at the individual level captures demographic characteristics, we argue that racism operates as a structural determinant at a broader, systemic level, influencing multiple aspects of food access in favelas. This was a point of discussion raised by the panel of experts, whose suggestions can be found in supplementary material 3. 

In Brazil, racism is a deeply rooted social structure that intersects with economic and social inequalities. It affects access to education, employment opportunities, income generation, and housing, influencing food security. These structural barriers are not solely individual experiences but are shaped by societal norms, institutional policies, and historical inequalities that disproportionately affect Black and marginalized populations living in favelas. For example, limited access to higher-paying jobs due to discrimination in the labor market directly impacts household income, restricting the ability to purchase adequate and nutritious food.

Moreover, geographic segregation often places predominantly Black communities in areas with poorer infrastructure and fewer supermarkets or markets that offer healthy food options. As such, racism functions beyond the individual level, embedding itself into policies, urban planning, and economic systems, thus shaping the macroenvironment and contributing to food deserts and food insecurity in these communities.

By including racism in the macroenvironment dimension, we emphasize its systemic and structural role in shaping food access in favelas. This perspective shifts the focus from individual demographic characteristics to understanding the broader societal forces that perpetuate inequalities in food systems, which is essential for creating comprehensive and just interventions.

As a follow-up study, employing ArcGIS to map and identify the favela's food environment would be interesting.

Thank you for your thoughtful feedback. We have already conducted some studies to see the distribution of formal food establishments in slum areas. The next step is to raise funds to conduct an on-site evaluation of the area to assess the distribution of informal establishments. 

Reviewer 2 Report

Comments and Suggestions for Authors

You did a great job in defining all the elements of this concept model in Table 1. I have the following comments and clarifications: 

Line 41: What types of buildings are the researchers referring to in “predominance of buildings”? Are these apartment complexes? Or makeshift structures?

Line 49: Are “dwellers” referring to favela residents?

Line 60: How are researchers defining “healthy food”

General comment: the word “healthy” is being used throughout the text. It would be helpful to define what is “healthy” as part of “healthy foods”

Line 61: Can you provide examples of ultra-processed and healthy food stores, especially in the context of Brazil?

Line 62: It would be helpful to briefly define the food environment when it is first mentioned. Then, it can be further described in the methods section (as it is currently).

Line 100: Which literature? Can you be specific?

Line 100: How many focus groups? How many individuals were in each focus group? How were participants selected? What was the demographic composition of those groups? 

Line 104: the concept of formal should be quickly defined here. What makes something formal, etc?  

General comment: Did the study occur in one specific favela, and if so, where or multiple sites? This is not clear from the text

Line 110: Which databases were searched to identify the bibliography? How many members of the research team were involved in the search? What were the inclusion/exclusion criteria? How were disputes about inclusion resolved? Did language matter in the inclusion/exclusion criteria?

Lines 130-136: Were the participants chosen randomly, or is this a convenience sample?

Figure 2: Is the arrow showing a one-direction? Is this implying that the relationships are not bidirectional?

Figure 2: The icons are hard to see and interpret. As of right now, I do not see their point.

Figure 2: The second box ends with a comma. Delete the comma

Figure 2: Decision-making levels column: Insert a space between the third and fourth points to make it consistent with the others

Table 1: The layout in the second column is odd. For example, in the column for the food literacy definition, “about” sits right in the middle of the second column.

Table 1: Is there a different term for “household structure”? This term seems to describe the household composition (e.g., number of family members, etc.)

Table 1: Informal food stores: What are “social innovations”

Table 1: Online food delivery platforms: What is ready-to-eat versus not ready-to-eat? Can you provide examples for both?

Line 332: It would be helpful to list which fields, professions or who specifically neglect the food environment

Author Response

Comments

Response

You did a great job in defining all the elements of this concept model in Table 1. I have the following comments and clarifications: 

Line 41: What types of buildings are the researchers referring to in “predominance of buildings”? Are these apartment complexes? Or makeshift structures?

Thank you for your careful reading of the work and your thoughtful feedback.

This phrase refers to buildings, road planning, and infrastructure that are often self-generated. A self-generated building is one that the dwellers make themselves—for example, wooden houses. 

Line 49: Are “dwellers” referring to favela residents?

Yes, a dweller is a person who lives in a city, town, or in a specific place (https://dictionary.cambridge.org/dictionary/english/dweller). Its synonyms are inhabitant or denizen. 

Line 60: How are researchers defining “healthy food”

According to the Brazilian Dietary Guide, healthy food is fresh and minimally processed. We describe this in the manuscript. 

General comment: the word “healthy” is being used throughout the text. It would be helpful to define what is “healthy” as part of “healthy foods”

As you suggested before, we described what it is the first time we mentioned it. 

Line 61: Can you provide examples of ultra-processed and healthy food stores, especially in the context of Brazil?

We described what the study we cited considered ultra-processed and healthy food stores. 

Line 62: It would be helpful to briefly define the food environment when it is first mentioned. Then, it can be further described in the methods section (as it is currently).

We described the food environment briefly in the introduction, as you suggested. 

Line 100: Which literature? Can you be specific?

We specify here that the literature we mention is about the food environment. 

Line 100: How many focus groups? How many individuals were in each focus group? How were participants selected? What was the demographic composition of those groups? 

We insert this information here briefly because they are well described in the article published. 

Line 104: the concept of formal should be quickly defined here. What makes something formal, etc? 

We briefly described what formal food stores are. 

General comment: Did the study occur in one specific favela, and if so, where or multiple sites? This is not clear from the text

The qualitative study involved favela dwellers from different cities in Brazil. The study of food stores took place in Belo Horizonte, Minas Gerais, a metropolis in Brazil. We include this information in the manuscript. 

Line 110: Which databases were searched to identify the bibliography? How many members of the research team were involved in the search? What were the inclusion/exclusion criteria? How were disputes about inclusion resolved? Did language matter in the inclusion/exclusion criteria?

We did not conduct a systematic review of the literature. We did a bibliography survey in PubMed, Scopus, Web of Science, and Scielo, without restricting the language, about the food environment and a favela (or slum) and about the food environment and conceptual or theoretical models. We rewrite a part of this paragraph to make the information clearer.

Lines 130-136: Were the participants chosen randomly, or is this a convenience sample?

We invited recognized experts and active civil society members who live in Brazilian favelas to participate in the panel. We rewrite the phrase to make it clearer. 

Figure 2: Is the arrow showing a one-direction? Is this implying that the relationships are not bidirectional?

The arrow was included to show that the decisional level interferes with all dimensions. However, with your comment, we agree that a bidirectional arrow is needed in this case to better show how this dimension could affect and interact with others. 

Figure 2: The icons are hard to see and interpret. As of right now, I do not see their point.

The icons represent some of the elements included in the dimension. In this version, it is not possible to see clearly, but we submitted the figure in high quality to the journal. 

Figure 2: The second box ends with a comma. Delete the comma

We deleted the comma.

Figure 2: Decision-making levels column: Insert a space between the third and fourth points to make it consistent with the others

We adjust the image. 

Table 1: The layout in the second column is odd. For example, in the column for the food literacy definition, “about” sits right in the middle of the second column.

We adjusted the table to fit in the pages and be better readable. 

Table 1: Is there a different term for “household structure”? This term seems to describe the household composition (e.g., number of family members, etc.)

We agree with your comment and have changed the term to Household kitchen equipment and utensils.

Table 1: Informal food stores: What are “social innovations”

Social innovations refer to new practices, strategies, or models adopted by these informal establishments to deal with social and economic challenges, such as new forms of payment (barter system) or selling in unconventional places. We added this information to the manuscript. 

Table 1: Online food delivery platforms: What is ready-to-eat versus not ready-to-eat? Can you provide examples for both?

Ready-to-eat food refers to fully prepared and cooked items that can be consumed immediately upon delivery, such as pizza or burger. On the other hand, not ready-to-eat food includes items that require preparation, cooking, or assembly before consumption, such as fresh vegetables or rice and oil.

Line 332: It would be helpful to list which fields, professions or who specifically neglect the food environment

When we mention that favelas are generally forgotten by public policies and studies on the food environment, we are pointing out that there are few or no policies and studies that specifically address this territory or the problems it faces in terms of access to food, for example. Studies on this topic are emerging after 2021 in Brazil, as well as a focus on food and nutritional security policies in these territories in a more proactive way.

Reviewer 3 Report

Comments and Suggestions for Authors

This is good study to provide current scenarios. Very well conceptualised and definitions of variables were provided. Theoretical conceptual framework is developed.

However, following points to be considered.

1. After forming the theoretical conceptual framework, it is important to identify the interaction between variables.

2. This need the quantitative analysis also. This component is missing in the model. Then it is needed to statistically model to identify interactions and most suitable variables.

If not these are limitations in this study and the objectives ro be adjusted for theoretical model.

Author Response

Comments

Response

This is good study to provide current scenarios. Very well conceptualised and definitions of variables were provided. Theoretical conceptual framework is developed.

However, following points to be considered.

1. After forming the theoretical conceptual framework, it is important to identify the interaction between variables.

2. This need the quantitative analysis also. This component is missing in the model. Then it is needed to statistically model to identify interactions and most suitable variables.

If not these are limitations in this study and the objectives ro be adjusted for theoretical model.

Thank you for your valuable feedback about the conceptual model. We want to clarify that this is a theoretical study with the primary goal of developing a conceptual model to understand the complex factors influencing food access in favelas. This work focuses on the qualitative relationships and dimensions that affect access to food, particularly in contexts of socio-spatial vulnerability.

Regarding the interaction between variables and the call for quantitative analysis, identifying and modeling these interactions statistically is an important next step for empirical research. However, this study does not aim to evaluate these relationships quantitatively. Our objective is to provide a foundation for future research, offering a structured framework to guide qualitative and quantitative investigations.

We agree that the lack of quantitative analysis is a limitation, as we have focused on the model's theoretical development. Nevertheless, this limitation aligns with the scope of our objectives, which are centered on conceptualizing the factors that influence food access rather than empirically testing their interactions at this stage.

Additionally, we recognize that developing or selecting appropriate instruments to measure all the elements presented in the model may be challenging. While no single instrument may capture the full complexity of the relationships discussed, combining existing tools could address more elements and provide a broader assessment. Future studies could explore integrating multiple instruments or developing new ones to capture the interactions between these variables better, combining qualitative and quantitative approaches.

In future research, we encourage using this model for statistical testing and exploring variable interactions through quantitative methods, potentially combining instruments that can address the broader scope of factors outlined in the model.